# From Conflict to Balance: Challenges for Dual-Earner Families Managing Technostress and Work Exhaustion in the Post-Pandemic Scenario

**DOI:** 10.3390/ijerph20085558

**Published:** 2023-04-18

**Authors:** Cataldo Giuliano Gemmano, Amelia Manuti, Sabrina Girardi, Caterina Balenzano

**Affiliations:** 1Department of Education, Psychology, Communication, University of Bari, 70121 Bari, Italy; giuliano.gemmano@uniba.it; 2Department of Political and Social Science, University of Bari, 70121 Bari, Italy; sabrina.girardi@uniba.it (S.G.); caterina.balenzano@uniba.it (C.B.)

**Keywords:** dual earners families, work–life balance and conflict, technostress, work exhaustion, new normal

## Abstract

Within the last three years, the COVID-19 pandemic outbreak has contributed to changing many aspects of individual and collective life. Focusing on professional life, the forced shift to remote working modalities, the consequent blurring of work–family (WF) boundaries, and the difficulties for parents in childrearing have significantly impacted family routines. These challenges have been more evident for some specific vulnerable categories of workers, such as dual-earner parents. Accordingly, the WF literature investigated the antecedents and outcomes of WF dynamics, highlighting positive and negative aspects of digital opportunities that may affect WF variables and their consequences on workers’ well-being. In view of the above, the present study aims to investigate the key role of WF conflict and WF balance in mediating the relationship between technostress and work exhaustion. Structural Equation Modeling (SEM) was used to examine direct and indirect relationships among technostress, WF conflict, WF balance, and work exhaustion. Respondents were 376 Italian workers, specifically dual-earner parents who have at least one child. Results and implications are discussed with specific reference to the organizational policies and interventions that could be developed to manage technostress and WF conflict, fostering individual and social adjustment to the new normal.

## 1. Introduction

Within the last three years, the COVID-19 pandemic has radically transformed many aspects of individual and collective life. Consequently, as a meaningful sphere of life, work experience has been concretely redefined by the necessity to adjust to new practices and environments and to learn and use (sometimes quite) new technologies. 

The widespread adoption of remote working brought important consequences for individuals in terms of relationships between work and family domains, and it arose the need for new strategies to keep the borders between personal and professional life [1]. Therefore, special attention was given to work–family (WF) balance [2] and its consequences for workers’ well-being during the pandemic [3]. Nonetheless, the relationship between work and family was considered one of the five emerging psycho-social risks in the labor market also before the COVID outbreak [4]. Therefore, in the current (post) pandemic scenario characterized by the effort to come back to a “new normal” and to recover from COVID-19, WF balance is becoming one of the most important focus of scientific investigations to understand whether the individual and organizational strategies developed during the emergency could be valid to protect the interface between work and family also compared to what experienced in the “old normal” [5].

Among the many changes introduced by the outbreak of the pandemic, the digital transformation is the one that is reaching a faster trajectory of development. Several studies investigated the consequences of digital opportunities, highlighting both positive and negative aspects that may impact individual well-being and productivity [6]. Some of the positive aspects of technologies that influence individual outcomes are related to the possibility of working from home, the opportunity for skill development, the accessibility to a wide variety of information, and the ability of digital communication to fill physical distances. During the pandemic, in particular, working from home was one of the most exploited work practices that showed several advantages since it offered more flexibility in managing daily tasks and gave workers the opportunity to take care of their families, to work during the most productive time, to avoid distractions from co-workers, to save expenses for travel, and to adapt the working environment to personal needs and preferences [7]. The combination of these factors leads to positive outcomes in terms of mental and physical well-being [8] and balance between work and family domains [7]. On the other hand, these positive aspects may hide some negative sides: the opportunity to take care of the family during working hours may lead to overwhelming demands, making the boundaries between work and family more blurred [9]. The possibility of avoiding distractions from co-workers may lead to social isolation and difficulties in collaborative work tasks [10]. 

Thus, despite their undeniable benefits, the rapid acceleration and widespread adoption of the so-called Information and Communication Technologies (ICT) also caused some negative consequences on individual experience, addressed as technostress [11]. Technostress is meant as the perceived sense of uneasiness induced by the pervasiveness, the complexity, and the invasion of technologies in personal, professional, and family life. Several studies revealed the impact of technostress on WF conflict, claiming that the use of ICT could blur the boundaries between work and family domains [12,13]. Moreover, technostress was investigated as a predictor of work exhaustion, since the strain caused by technology may deplete workers’ resources, limiting their capacity to respond to job demands and leading to feelings of exhaustion [14,15]. Although previous studies highlighted the relationships between technostress, WF conflict, and work exhaustion [16], the literature concerning the intervening role of WF balance in this “negative” chain is scarce [17]. 

In light of these premises, the current study was guided by the curiosity to explore these relationships with specific reference to a vulnerable target of workers challenged by some of the consequences of the pandemic: dual-earner parents. Evidently, because of their double role as workers and parents, dual-earners might have been double exposed to the several challenges brought about by the emergence: for those who have been forced to work in presence, they might have experienced the fear to be infected, for those who had a precarious job and/or a job linked to entertainment and social recreation (e.g., fitness, self-care, food and beverage, theatres and cinemas, etc.), they might have experienced the fear to lose their job, for those who have been forced to abruptly shift to remote working, they might have experienced techno-stress and the perception of not being sufficiently ready to this new, complex and pervasive working modality. This emotional load might have impacted the management of WF balance if certainly dual-earner parents have been forced to hardly re-organize their working modalities as well as their family routines. Accordingly, the outbreak of the pandemic has forced dual-earner families engaged in childrearing to adjust to the concurrent deprivation of any kind of social support, be it the one granted by the informal network of their family of origin as well as the formal one provided by educational and social services. This experience has turned into work-related stress, family stress, and burnout [18], especially for women, who at least in Italy still bear the main responsibility in households [19]. 

However, in facing these difficulties, following the job resources/demands model, which postulates that strain and stress might derive from a perceived imbalance between the demands on the individual and the resources he or she has to deal with those demands [20], WF balance—meant as “the individual perception that work and non-work activities are compatible and promote growth in accordance with an individual’s current life priorities” [21] (p. 326)—could be conceived as a precious personal resource of resilience in this scenario.

In view of the above, the present study aims at investigating the key role of WF balance in mediating the relationship between WF conflict and work exhaustion. The intended contribution of the study is to shed light on the importance of enhancing WF balance in the “new normal”, as this is the main protective factor that could contribute to buffering the impact of technostress and WF conflict on dual-earner parents’ work exhaustion. Therefore, individual and organizational interventions aimed at developing strategies to strengthen WF balance may help working parents face the negative consequences of the conflict between work and family domains, finally improving their well-being.

The present contribution presents the following sections: Theoretical Background to examine the literature concerning the relationships between technostress, WF conflict, WF balance, and work exhaustion that supports the study’s hypotheses; Material and Methods to show information about the procedure, participants, measures, and analyses; Results to present the output of preliminary analyses and Structural Equation Modeling (SEM); Discussion and Conclusions to interpret the findings and highlight theoretical and practical implications; and Strengths, Limitations, and Perspectives to further examine the study’s characteristics and propose future research.

## 2. Theoretical Background

### 2.1. WF Conflict, WF Balance, and Work Exhaustion

Over the past few decades, the dynamic relationship between work and family domains has gained a growing interest in the scientific literature [22,23], because of the increase in dual-earner families, the diffused possibility/constraint of working from home, and the pervasive use of technology that concretely risks blurring the boundaries between family and work life [24,25,26]. Scholars investigated these phenomena to understand their consequences for workers’ and families’ well-being, adopting different perspectives to conceptualize WF dynamics in positive or negative terms: some scholars focused on the construct of WF balance [27], some on WF enrichment [28], some on WF spillover [29], and some others on WF conflict [30].

Among these different conceptualizations of the WF constructs, the interrelations between work and family have been largely explored, focusing on their negative aspects to investigate the extent to which one domain of life may weaken the other [31,32]. A plethora of studies focused on WF conflict, which was considered for years as the most representative indicator of complex inter-role dynamics between work and family [22]. WF conflict is defined as a form of inter-role conflict in which work, and family roles are incompatible due to the pressure of the different domains [20]. The definition is consistent with the role strain hypothesis [33], addressing the difficulties of participating in different social roles, which may interfere with other life domains and have conflicting or overlapping demands. Carlson and colleagues [34] investigated the different types of pressure that could originate from work and family domains, claiming that WF conflict consists of time-based, strain-based, and behavior-based pressures that cause inter-role conflicts. Accordingly, the presence and engagement in each of the specific domains require time, which is a limited resource, forcing individuals to choose whether to devote it to the work domain or the family domain. Furthermore, performance in each domain involves strain that could negatively affect participation in the other one. Likewise, each domain demands specific behaviors that sometimes are incompatible with the behavioral expectations posed by the other domain. All these pressures may originate in each role and affect the other; therefore, WF conflict has been investigated in both directions: work interfering with family and family interfering with work [35]. The first direction is determined by work demands, such as long working hours, work overload, and work-related stress, which may have negative consequences for the family role [36,37]. The other direction is determined by family demands, such as responsibilities toward family members, which require time and energy and may weaken the worker role [38]. 

Although the WF literature has been largely dominated by a negative perspective focusing on WF conflict [39,40], recently a positive perspective was developed, conceptualizing WF balance as a holistic construct concerning the compatibility between work and family roles [41]. Originally, WF balance was considered the absence of conflict or the presence of low levels of inter-role conflicts and high levels of inter-role facilitations [35]. However, this conceptualization failed in capturing the wide range of experiences regarding WF dynamics and in considering the possibility that individuals could adopt strategies inspired by WF balance even in conditions of high levels of WF conflict. Therefore, further research recognized WF balance as a distinct construct defined as the “accomplishment of role-related expectations that are negotiated and shared between an individual and his or her role-related partners in the work and family domains” [27] (p. 458). This definition highlights the importance of the presence of other meaningful people in both domains for the co-construction of role expectations and for their accomplishment. The conceptual distinction between WF conflict and WF balance was investigated in validation studies, which confirmed the discriminant validity between measures of the two constructs and highlighted a negative relationship between them [42,43,44]. The nature of the relationship was further explored in review and research articles proposing WF balance as an outcome of WF conflict and confirming a negative effect of WF conflict on WF balance [17,45,46,47,48]. Building from this association, a few studies explored the mediating role of WF balance in the relationships between WF conflict and different outcomes in the work and family domains, such as job satisfaction [49], family satisfaction [50], and life satisfaction [51].

Most of these mediation studies explored the extent to which the effect of WF conflict mediated by WF balance may impact positive outcomes. However, very few studies have investigated a similar mediation path with negative outcomes, such as work exhaustion [52]. Work exhaustion is a core component of burnout and refers to feelings of being depleted of emotional and physical resources while managing working situations [53]. It reduces individual initiative, causes work strain, and limits workers’ capacity to adequately complete tasks. Work exhaustion may be a consequence of recurrent emotional and physical stress caused by high-demanding life situations [54]. Therefore, it was investigated as an outcome of both WF conflict [55] and WF balance [56] since the dynamics between work and family domains may represent emotional demands. Consistently with the COR theory [57], resource depletion associated with inter-role conflicts and low levels of WF balance may cause feelings of exhaustion and frustration at work. Although WF literature thoroughly investigated the direct relationships between the two WF constructs and work exhaustion, to our knowledge, no studies explored the mediating role of WF balance in the relationship between WF conflict and work exhaustion. The present study addressed this gap, claiming that WF conflict led to work exhaustion because it weakens WF balance, which is a protective factor against exhaustion. Likewise, a low level of inter-role conflict could help the achievement of a WF balance by preventing exhaustion. Therefore, we expected that:

**H1.** 
*Work–family conflict is negatively related to work–family balance.*


**H2.** 
*Work–family balance is negatively related to work exhaustion.*


**H3.** 
*The relationship between work–family conflict and work exhaustion is mediated by work–family balance.*


### 2.2. Technostress and Work–Family Dynamics

Scholars have widely explored the antecedents of WF constructs to understand which are the personal or contextual factors that could influence the relationship between work and family domains [40,58,59,60]. Among these factors, technostress has been the focus of scientific research in recent years due to the increase of ICT usage in all fields and the urgent necessity of technology caused by the pandemic of working from home and being connected to others [61,62]. Technostress is defined as the stress experienced because of the use of technologies, often caused by an overload of information, the demands of multitasking, difficulties in solving technical problems, the necessity of being constantly connected, and the urgency of updating technical skills [11]. Some studies tried to identify the factors that contribute to technostress by distinguishing the following categories of “technostress creators” [63]: techno-overload, referred to the overload caused by the great amounts of information and stimuli of ICT; techno-invasion, related to the intrusion of technology into daily activities with the consequence of the absence of boundaries between different life areas; techno-complexity, which concerns the excessive complexity of technologies that lead to feelings of inadequacy in front of digital difficulties; techno-insecurity, referred to the fear of losing one’s own job because technologies could replace working activities; and techno-uncertainty, related to the unpredictable changes due to technological developments that require constant updates of digital knowledge.

Technostress creators were investigated in the Italian context during the pandemic with specific reference to the components of overload, invasion, and complexity, which were considered predictors of WF conflict during remote working [16]. This finding is consistent with a growing body of literature suggesting that the negative effects of technostress creators may spill over into the family domain [64,65,66]. Many studies explored the influence of technostress on WF conflict, claiming that the use of ICT for working tasks may contribute to making the boundaries between work and family more blurred, leading to inter-role conflicts [12,13]. In fact, the invasion of technology brings work home at every hour and distracts the individual from family responsibilities [67]. Similarly, the ICT overload and complexity require attentional and time resources that may deplete workers’ energy, making it difficult to adequately participate in family life and meet work and family expectations [65].

Beyond the consequences on WF dynamics, the stressors of technology may lead to exhaustion, depletion, and frustration in the work domain [14]. Strain caused by technology may deplete workers’ resources, limiting their capacity to respond to job demands and leading to feelings of inadequacy and exhaustion. In recent years, the influence of technostress on work exhaustion and burnout was explored to identify the potential risks of technology for workers’ well-being [15]. Many studies confirmed that technostress may have direct and indirect effects on work exhaustion, exploring also the mediating and moderating roles of intervening variables [68,69,70]. Given the influence of WF constructs on work exhaustion [71], some studies attempted to highlight the path from technostress reach exhaustion to the mediation of WF conflict [16]. Nevertheless, none of them explored the extent to which WF balance may intervene in the path between conflict and exhaustion.

Therefore, the present study aimed to fill this gap, exploring the mediating roles of both WF conflict and WF balance in the relationship between technostress and work exhaustion. Specifically, we propose a research model (see Figure 1) that sees technostress as a predictor of WF conflict, which in turn influences WF balance, which is a protective factor against work exhaustion. Therefore, along with the former three hypotheses, we expected that:

**H4.** 
*Technostress is positively related to WF conflict.*


**H5.** 
*The relationship between technostress and work exhaustion is mediated by WF conflict and WF balance.*


## 3. Materials and Methods

### 3.1. Procedure and Participants

A convenience sample of working parents was recruited from July to September 2022 by the research team that supervised the online completion of questionnaires. Considering that small and medium-sized enterprises (SME) account for the majority of Italian firms, the sample was recruited at several different organizations. A short description of the research aims followed by an invitation to participate voluntarily and anonymously in the study was presented to participants. The study observed the Helsinki Declaration and the prescriptions of the General Data Protection European Regulation (EU n. 2016/679). Respondents were 376 Italian dual-earner parents who have at least one child. After data cleaning, the final sample consisted of 361 working parents because 15 invalid or missing records were removed (42% were women and 58% were men). At the time of data collection, all the participants worked in the same room. Participants’ mean age was 41.56 years (*SD* = 8.55). As for their education, 32% of participants had a bachelor’s degree, 46% were graduates of secondary school, and 22% had lower educational levels. Most of the sample had a full-time job (73%) in the private (80%) or public (20%) sectors. As for family income bracket, 13% of participants fell into the lowest category (lower than EUR 15,000), 53% into the medium-low category (EUR 16,000–33,000), 29% into the medium-high category (EUR 34,000–55,000), and 6% into the highest category (over EUR 55,000).

### 3.2. Measures

To explore the hypothesized model, participants were invited to fill out an online questionnaire composed of a section dedicated to the collection of socio-demographic information and another one made up of some psychometrically validated measures used to assess the following variables: technostress, work–family conflict, work–family balance, and work exhaustion.

Technostress. The Technostress Creators Scale [11], validated in Italy by Molino and colleagues [16], was used to assess this variable. The instrument aims to investigate the stress caused by technology. It consists of eleven items for three sub-scales: four items of techno-overload, describing the compelling load of working faster and longer because of technology (e.g., “I am forced by technology to do more work than I can handle”); three items of techno-invasion, related to the perception that technologies may invade all areas of life (e.g., “I feel my personal life is being invaded by this technology”); and four items of techno-complexity, referring to the feeling of inadequacy in using technologies (e.g., “I often find it too complex for me to understand and use new technologies”). Participants were invited to use a Likert scale from 1 (strongly disagree) to 5 (strongly agree). In our sample, Cronbach’s α for overload, invasion, and complexity was 0.94, 0.79, and 0.92, respectively;Work–family conflict. A short version of Carlson, Kacmar, and Williams’s [34] scale, validated by Matthews and colleagues [72], was used to assess this variable. The scale consists of three items describing time-based, strain-based, and behavior-based conflicts between work and family domains (e.g., “I have to miss family activities due to the amount of time I must spend on work responsibilities”). Items were rated on a 5-point Likert scale ranging from 1 (strongly disagree) to 5 (strongly agree). In the present study, Cronbach’s α was 0.94;Work–family balance. This variable was investigated by the Italian version of the Work–Family Balance scale [42], validated by Landolfi and Lo Presti [44]. The scale assesses the extent to which an individual can accomplish work and family role expectations. It consists of six items on a 5-point Likert scale ranging from 1 (strongly disagree) to 5 (strongly agree) (e.g., “I am able to negotiate and accomplish what is expected of me at work and in my family”). In our study, Cronbach’s α was 0.93;Work exhaustion. This variable was assessed through five items taken from the emotional exhaustion scale of the Maslach Burnout Inventory General Survey [73]. It concerns feelings of being emotionally drained, frustrated, and exhausted by work. Participants were asked to express the occurrence of each item in their ordinary work experience using a 6-point scale from 1 (never) to 6 (everyday) (e.g., “I feel emotionally drained from my work”). In the present study, Cronbach’s α was 0.87;Control variables. Age and gender were added to the model as control variables since they could have a role in affecting work–family constructs and work exhaustion, as stated in previous studies [74,75,76].

### 3.3. Data Analysis

Preliminary analyses were performed before testing the model, including the exploration of means, standard deviations, normality of distributions, reliability measures, and Pearson correlations between the study variables. The validity of the measures was evaluated by conducting a preliminary Confirmatory Factor Analysis (CFA) aimed at testing the measurement model. The Fornell–Larker criterion [77] was used to confirm discriminant validity. Structural Equation Modeling (SEM) was used to analyze the fit of our hypothesized structural model and evaluate direct and indirect relationships between the latent variables. 

The average scores of the items on each scale were used to explore descriptive statistics and correlations between the variables of interest. The raw items were used as observed variables of the respective latent construct in the preliminary CFA and SEM. Although observed variables have skewness and kurtosis values ≤ |1.00|, suggesting univariate normal distributions [78], the multivariate normality assumption is not met since the Henze–Zirkler test [79] shows a statistically significant value of 1.10, *p* < 0.001. Therefore, the CFA and SEM analyses were conducted using the maximum likelihood method of estimation with robust standard errors (MLR). All analyses were carried out using the R software [80] and the R package *Lavaan* [81]. The significance of the indirect effects was investigated through a bootstrapping procedure that extracted 5.000 new samples from the original one. The goodness of fit of models was evaluated considering the chi-squared statistic and additional pragmatic fit indices [82], namely the Comparative Fit Index (CFI), the Tucker Lewis Index (TLI), the standardized root mean square residual (SRMR), and the root mean square error of approximation (RMSEA). Values of CFI and TLI greater than 0.90 and values of SRMR and RMSEA lower than 0.08 suggest a good fit of the model.

## 4. Results

### 4.1. Preliminary Analyses

Table 1 shows means, standard deviations, Cronbach’s alpha coefficients, and Pearson correlations between variable scores and control variables (i.e., age and gender). All correlations between the variables of interest reach significance and take the expected directions, showing positive bivariate relationships between technostress, WF conflict, and work exhaustion and negative relationships between WF balance and the other main variables. Age is positively related to the total score of technostress. Gender (recoded into 0 for men and 1 for women) is negatively related to WF conflict, showing that men perceived a higher level of conflict than women. On the other hand, gender has non-significant correlations with the other variables of the study’s interest, suggesting that there are no differences between women and men in the perceptions of technostress, WF balance, and work exhaustion. Internal consistency of scales is acceptable since Cronbach’s alpha values ranged from 0.79 to 0.94.

The hypothesized model aims to investigate the mediating role of WF conflict and WF balance in the relationship between technostress and work exhaustion. In the measurement model, technostress is defined as a second-order variable measured by the first-order factors of techno-overload, techno-invasion, and techno-complexity. The adoption of a second-order variable aimed to develop a more parsimonious model and reduce the risk of multicollinearity between independent variables. The other variables of interest are defined as first-order factors, each measured by its own set of items. A preliminary CFA was performed to evaluate the measurement model, highlighting a good fit with acceptable values of pragmatic indices: χ^2^ (266) = 590.88, *p* < 0.001, CFI = 0.95, TLI = 0.95, RMSEA = 0.06, SRMR = 0.06, and significant factor loadings for all the indicators of each latent variable. The sizes of the standardized factor loadings are between 0.82 and 0.94 for techno-overload; between 0.70 and 0.78 for techno-invasion; between 0.76 and 0.93 for techno-complexity; between 0.87 and 0.97 for WF conflict; between 0.76 and 0.89 for WF balance; and between 0.65 and 0.85 for work exhaustion. The correlations between latent variables range between |0.27| and |0.39| and do not suggest collinear relationships. Since the measures of different constructs were taken together through a self-report questionnaire, we used Harman’s single factor as a diagnostic technique to check if the common method bias could have been a problem [83]. The total variance extracted by one factor was 23.57%, which is lower than the cutoff value of 50% that is suggested in the literature for Harman’s test [84]. Moreover, the CFA addressing a measurement model with one factor shows a bad fit: χ^2^ (275) = 4939.31, *p* < 0.001; CFI = 0.31, TLI = 0.25, RMSEA = 0.22, SRMR = 0.21, indicating that a single factor does not account for all the covariances among the indicators.

Since WF conflict and WF balance have similar yet different meanings and both refer to perceptions regarding the relationship between work life and family life, the discriminant validity was tested to confirm the conceptual distinction between the two constructs. The Fornell–Larcker criterion [77] was applied to verify that the square root of the average variance extracted (AVE) values of each construct (i.e., 0.92 for WF conflict and 0.83 for WF balance) are higher than the correlation in absolute value between the latent variables (f = |−0.27|). These results were consistent with previous studies [42,85].

### 4.2. SEM Analysis

The structural equation model investigates direct and indirect relationships between technostress and work exhaustion through the mediation of WF conflict and WF balance, controlling for age and gender. The hypothesized model also explores the mediating effect of WF balance in the relationship between WF conflict and work exhaustion. Considering the multiple predictors of work exhaustion, multicollinearity was checked by examining Variance Inflation Factors (VIF), which showed acceptable values of 1.22 for technostress, 1.24 for WF conflict, and 1.11 for WF balance, indicating that multicollinearity is not an issue. The goodness of fit of the model was acceptable: χ^2^ (310) = 652.21, *p* < 0.001, CFI = 0.95, TLI = 0.94, RMSEA = 0.05, SRMR = 0.06.

Figure 2 shows the structural model with standardized direct effects. As regards direct relationships, technostress has a significant positive effect on WF conflict (β = 0.38, *p* < 0.001), a significant negative effect on WF balance (β = −0.15, *p* < 0.05), and a non-significant direct effect on work exhaustion (β = 0.08, *p* > 0.05). WF conflict shows a significant negative effect on WF balance (β = −0.25, *p* < 0.001) and a non-significant direct effect on work exhaustion (β = 0.12, *p* > 0.05). Finally, the negative direct path coefficient from WF balance to work exhaustion reached significance (β = −0.30, *p* < 0.001). As regards control variables, age does not show significant effects, and gender has significant negative effects on WF conflict (β = −0.29, *p* < 0.001) and WF balance (β = −0.13, *p* < 0.05). The total effects of technostress (β = 0.20, *p* < 0.01) and WF conflict (β = 0.19, *p* < 0.01) on work exhaustion are both positive and significant. The significance of the indirect effects was tested by using the bootstrapping method. The indirect effect of technostress on work exhaustion, mediated by WF conflict and WF balance, is statistically significant (B = 0.03, bootstrapped 95% CI = (0.01; 0.05)). Similarly, the indirect effect of WF conflict on work exhaustion via WF balance reaches significance (B = 0.06, bootstrapped 95% CI = (0.03; 0.10)).

Overall results show that people with high levels of technostress are more likely to perceive more conflict and less balance between work and family life. From a complementary perspective, working parents with low levels of technostress show lower levels of WF conflict and higher levels of WF balance. These relationships suggest that the variability of technostress impacts WF dynamics in different ways, since positive experiences with technology (i.e., low levels of technostress) may help improve the relationship between work and family domains, whereas negative experiences with technology (i.e., high levels of technostress) may exacerbate WF conflict and weaken WF balance. Furthermore, the sequential mediation path indicates that technostress is indirectly associated with work exhaustion because of WF variables. It could also be observed that the relationship between WF conflict and work exhaustion is mediated by WF balance, suggesting that conflicts may have an indirect effect on exhaustion because they reduce the balance that could be a protection factor from exhaustion. 

## 5. Discussion and Conclusions

The purpose of the study was to investigate the intervening role of WF balance in the relationship between WF conflict and work exhaustion in the “new normal” scenario after the COVID-19 pandemic. Moreover, the study explored the extent to which the effect of technostress on work exhaustion was mediated by both WF conflict and WF balance.

The first two hypotheses were confirmed: results showed that WF conflict was negatively related to WF balance (H1), and WF balance was negatively related to work exhaustion (H2). Therefore, the present study sustained WF literature highlighting the negative influence of inter-role conflict on the balance between work and family domains [17,48] and the protective role of WF balance from work exhaustion [3,56]. In addition, factor analyses contributed to confirming the conceptual difference between WF balance and WF conflict, consistent with previous studies [42,44,86]. One of the main results of the present study highlighted the mediating effect of WF balance in the relationship between WF conflict and work exhaustion (H3). This evidence helps shed light on the consequences of inter-role conflict on exhaustion, showing that WF balance plays a key role in that relationship. WF conflict may impact work exhaustion by decreasing working parents’ ability to keep work and family responsibilities and expectations in balance. The mediation role of WF balance between WF conflict and work exhaustion represents an original contribution to WF literature, which was dominated by studies focused on the effect of just one WF construct on exhaustion [55,56] or studies exploring both the effects of WF conflict and WF balance on positive outcomes [50,51]. The present study intended to reconcile a negative and a positive perspective about the relationship between work and family domains, showing the association between WF conflict and WF balance after demonstrating their conceptual uniqueness. Furthermore, it shows how the relationship between the two WF constructs may contribute to explaining the variance of work exhaustion, a negative indicator of well-being.

In regards to technostress, results confirmed its direct relationship with WF conflict (H4) and indirect relationship with work exhaustion through the mediation of both WF conflict and WF balance (H5). The impact of technostress on WF dynamics and exhaustion underlined a widespread problem of the last few decades related to ICT, its overload, invasion, and complexity [12,16]. Consistent with previous research [13,61,62,63,64,65,66], the present study claims that technostress creators may interfere in the relationship between the work and family spheres, contributing to the blurring of the boundaries between the two domains and causing inter-role conflicts. For example, the excessive use of technologies for working tasks outside of working hours may deprive the family domain of energy, attention, and time, leading to difficulties in complying with both work and family demands. This has more serious implications for dual-earner parents, who are both simultaneously engaged in managing professional and family engagements. 

Moreover, the evidence concerning the mediating effect of both WF conflict and balance in the relationship between technostress and work exhaustion contributes to the enrichment of the literature. Although researchers have highlighted the effect of technostress on work exhaustion [14,15] and the mediation of WF conflict [16], as far as we know, no study has explored the intervening role of WF balance in this path. The focus on the balance between work and family domains contributes to explaining how technostress may impact work exhaustion. Given the influence of technostress on the variance of WF conflict, the present study highlighted that WF conflict’s variability affects work exhaustion because it determines the levels of WF balance, which is a protective factor from exhaustion. In other words, high levels of technostress lead to WF conflict, which in turn decreases WF balance, with negative consequences for work exhaustion.

Overall, the study gave also a contribution to the theoretical development in the field at least in four directions: (a) it investigated direct and indirect effects of technostress and WF constructs on work exhaustion; (b) it confirmed the theoretical distinction between WF conflict and WF balance; (c) it explained that WF conflict influences work exhaustion because it affects WF balance, considered a protective factor from exhaustion; and (d) it explored the mediating role of both WF conflict and balance in the relationship between technostress and work exhaustion.

Practical implications of the study could be directly linked to the emerging “new normal” scenario mentioned above. First, the issues raised by the diffusion of ICT and consequently the need to manage WF dynamics are still and will likely be relevant for dual-earner parents’ well-being in the future post-pandemic context. Specifically, technology may exasperate the relationship between work and family, leading to inter-role conflict and higher levels of work exhaustion. On the other hand, the present study highlighted the significant role of WF balance, which could break the negative chain created by the loop of technostress–conflict–exhaustion. Therefore, empowering strategies to accomplish socially negotiated role-related responsibilities and expectations may help working parents to decrease their level of exhaustion, damping the negative influence of technology and WF conflict. Accordingly, organizations are called to develop policies and interventions to monitor and manage technostress and WF conflict. Human Resources Management professionals could think about the possibility of implementing plans for flexible work designs [87], capitalizing on the good practices positively experienced during the long pandemic, as working-from-home plans, management by objectives, or time flexibility [5]. Moreover, organizations could involve employees in developing customized practices through dedicated needs analysis aimed at collecting subjective orientations for specific work environment options and/or working hours shifts, considering that some individuals may prefer to work in presence while others would be more productive at home to better balance work and family roles. Dedicated stress-management interventions could be addressed to reduce technostress and help working parents manage conflict with their families and exhaustion at work. For example, training sessions aimed at managing the complexity of the use of some specific technologies may reduce feelings of inadequacy in front of digital difficulties, with positive consequences on workers’ performance and individual well-being [21]. Likewise, interventions addressed to enhance WF dynamics could develop individual resources and competencies that can be used in both work and family domains (e.g., through specific job crafting interventions). Finally, interventions involving employees should also be complemented with interventions addressed to supervisors and HRM professionals to help them manage the difficulties of the “new normal”, to exercise an empowering and positive leadership style, and to support and convey a people-based and change-oriented culture. 

### Strengths, Limitations, and Perspectives

The current study has several important strengths. First, to our knowledge, this is the first study in the Italian context that used a psycho-sociological framework to explore the effects of working parents’ technostress on their family and work lives, as well as to test the mediating role of WF balance in the relationship between WF conflict and work exhaustion. Indeed, only recently have scholars focused on the negative consequences of excessive use of technologies [88], thus research examining the impact of working parents’ “overload” and their technostress on family and work life is lacking [89,90]. We think, instead, that since the impact of technostress on work–family balance represents a social problem [91], it is crucial to combine the psycho-occupational and the sociological approaches. Therefore, we examined the impact of technostress both on family conflict and balance, respectively used as negative and positive indicators of family well-being, and on work exhaustion, used as a negative indicator of working parents’ well-being. While most studies have focused on WF conflict or WF balance, we have measured both variables to explore the unique contribution that each indicator has on working parents’ work exhaustion. Lastly, data were analyzed through Structural Equation Modeling (SEM), a data analytic strategy useful to test both the direct and indirect effects of independent variables on the outcome. 

However, from a methodological perspective, some limitations of the present study should be mentioned. First, the sampling procedure is not probabilistic, and the sample is not representative of all the Italian regions, which means that our findings cannot be generalized. Considering the study’s inclusion criteria, the findings are specifically limited to participants with the following characteristics: they have one or more children, and they work in person. Consequently, the generalizability of the present study is limited because different results may arise in samples of workers with different characteristics. For example, remote workers may differently experience the stress caused by technology and its relationship with WF dynamics. Additionally, families without children may adopt diverse strategies to balance work and family life, showing higher or lower levels of WF variables that may show different relationships with technostress and work exhaustion. Furthermore, because of selection bias, it is possible that the working parents who participated in the study were significantly more motivated and/or more satisfied with their parental and/or working roles as working parents than those who did not. Thus, a larger random sample belonging to different regions in Italy, but also to other countries, would be ideal to be considered for studying the WF dynamics in (post) pandemic scenarios. Second, a relevant limitation of our study is that all the latent variables used in the SEM (i.e., technostress, WF conflict, WF balance, and work exhaustion) were based on self-report measures, which may have caused shared method variance and an overestimation of some of the associations. Even if we used validated scales, the use of only self-report measures requires that caution should be exercised when the data are being interpreted. Therefore, future research should include other types of measurements and involve multiple informants (e.g., external observers), combining qualitative and quantitative methods to collect both data on the subjective experience of working parents and more objective indicators in the subjects of work-related stress, work–family dynamics, and burnout. Finally, since this study was conducted using a cross-sectional design, it is not possible to argue that the observed mediating effect is necessarily causal, also because other factors not included in our model may have influenced the path. Future research may consider a longitudinal design to further investigate possible causal relationships among these constructs, as well as explore the role of other possible intervening factors in the model. 

In this regard, the role of family and sociodemographic variables could be further investigated to be able to give a more comprehensive picture of determinants of working parents’ burnout, especially the relationships among technostress, WF conflict, WF balance, and work exhaustion. Consistently with the literature highlighting the role of the support of the partner in family balances [92], we suggest investigating the contribution of the partner, for example in managing domestic tasks and taking care of children, in moderating the effect of technostress on WF conflict and balance. Moreover, since having preschoolers could lead to higher levels of burnout and WF conflict and/or lower levels of WF balance compared to having older children [93,94], data on the number and ages of children in the family should be considered in further studies. Finally, based on the evidence from literature [95], also the level of childcare support, particularly from grandparents, should be measured as a protective factor able to reduce WF conflict and burnout. 

More generally, as the COVID-19 pandemic raises new and specific challenges to WF balance [96], more complete and robust studies will be needed to better investigate family dynamics in the (post) pandemic scenario. In terms of perspectives, by adopting the inequality approach in family context [97], scholars would better explore how socio-demographic characteristics, and especially gender, influence the relationship between technostress, WF conflict, and WF balance. This line of research is particularly interesting in Mediterranean European countries such as Italy, where new facets of gender inequality related to the effects of the use of ICT on working and family life, and especially on work–family balance, are emerging [91,98]. Overall, since metanalytic evidence on work–family boundaries highlighted that gender asymmetries persist [99], a broader sociological approach that is more attentive to cultural factors could be helpful to investigate the influence of such asymmetries on gender inequalities concerning both informal work (domestic and childrearing tasks) and career’s resources and perspectives.

However, despite some shortcomings and beyond future broader studies, the present study has some interesting implications for social policies. Taken together, our results underscore that the numerous demands related to the multiple roles of working parents make more challenging the pursuit of work and family well-being, highlighting thus the need to plan new WF balance programs, interventions, and instruments to protect working parents from the negative consequences of technostress on WF dynamics. As WF conflict is a worsening social phenomenon in the aftermath of the pandemic, public and private stakeholders should cooperate to redesign WF balance policies in this post (pandemic) scenario, that is, redefine welfare systems to make them resilient in coping with the needs of balancing work and family life in dual-earner families [100]. Especially in a familial society such as Italy [101], new methods to buffer the different dimensions of technostress and their repercussions on working parents’ well-being, and especially mothers, should be tried. This recommendation concerns all the workers who in their work-day lives use digital communication methods, which can lead to stress and burnout, but especially remote working parents, who may not have the social support typically provided by the workplace and for whom the work could easily spill over into free time, causing negative consequences on well-being and WF balance [102]. 

## Figures and Tables

**Figure 1 ijerph-20-05558-f001:**
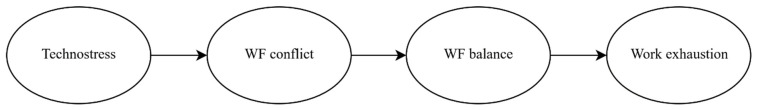
The research model. *Note*. Arrows refer to the hypothesized direct effects.

**Figure 2 ijerph-20-05558-f002:**
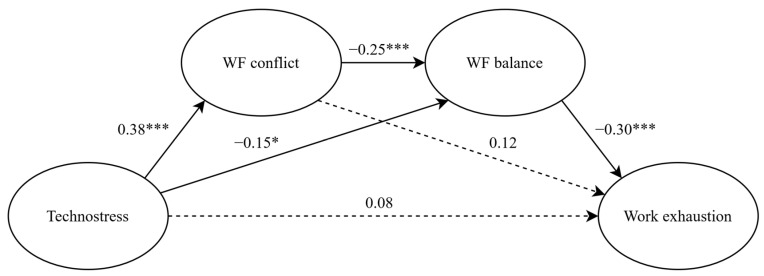
Structural model with standardized direct effects. *Note*. *** *p* < 0.001, * *p* < 0.05. Discontinuous lines indicate non-significant parameters. Indicators and control variables are not shown for the sake of clarity.

**Table 1 ijerph-20-05558-t001:** Correlations between variables scores and control variables.

	M	SD	Skew	Kurt	1	2	3	4	5	6
1. Technostress (1–5)	2.45	0.97	0.55	−0.26	—					
2. WF conflict (1–5)	3.34	1.25	−0.36	−0.90	0.30 ***	—				
3. WF balance (1–5)	4.16	0.73	−0.72	0.10	−0.19 ***	−0.28 ***	—			
4. Work exhaustion (1–6)	2.39	1.03	0.85	0.54	0.15 **	0.21 ***	−0.33 ***	—		
5. Age	41.56	8.55	0.16	−0.49	0.19 ***	0.05	−0.08	0.05	—	
6. Gender	—	—	—	—	−0.06	−0.31 ***	−0.02	−0.02	−0.15 **	—

Note. M—mean; SD—standard deviation; Skew—skewness; Kurt—kurtosis; WF—work–family. ** *p* < 0.01; *** *p* < 0.001.

## Data Availability

The data presented in this study are available on request from the corresponding author. The data are not publicly available due to privacy restrictions.

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
