# Peer review of "From Conflict to Balance: Challenges for Dual-Earner Families Managing Technostress and Work Exhaustion in the Post-Pandemic Scenario"

_ijerph, 2023, doi:10.3390/ijerph20085558_

Round 1

Reviewer 1 Report

Dear autghors,

I find your paper well-written and I think that the theme is very interesting for new normality - post Covid 19 era, in which many of pandemic solutions related to job are even more in use.

I read your paper and I think that is worth of reading and publication. I have just few suggestions.

1. Please, add that you used SEM in the abstract - it is going to be more interesting for researchers... 

2. Add the structure of the paper at the end of the introduction section.

3. Harman’s single factor was calculated, what was the exact value?

4. What are the VIFs for the variables?

Author Response

Reviewer 1

Dear authors,

I find your paper well-written and I think that the theme is very interesting for new normality - post Covid 19 era, in which many of pandemic solutions related to job are even more in use.

  1. Please, add that you used SEM in the abstract - it is going to be more interesting for researchers... 

Response 1. Thank you for pointing it out. We added the use of SEM in the abstract.

  1. Add the structure of the paper at the end of the introduction section.

Response 2. Thank you for this suggestion. We added the structure of the paper at the end of the introduction.

  1. Harman’s single factor was calculated, what was the exact value?

Response 3. Thank you. We added details for this analysis.

  1. What are the VIFs for the variables?

Response 4. Thank you. We added VIF values in the Results section.

Reviewer 2 Report

Thank you for the opportunity to review this timely manuscript.  I believe that it would make a nice addition to the literature on this topic.  I provide a few suggestions and questions below:

p. 2: What does "social recreation" refer to in the context of this sentence?

The authors appear to use "work exhaustion" and "emotional exhaustion" interchangeably when discussing this concept within their theoretical model, but readers may find that confusing.  

p. 5: Were survey respondents drawn from the same workplace or company?   A "rich mix" of companies?  Were they recruited from a particular sector of the economy?  Did they work in fields/jobs that could be characterized by high levels of technostress?  Did they work in the private or public sector?  More information about their workplaces (and job roles, for that matter) would be helpful to contextualize these findings within the broader literature.    

In terms of potential generalizability, the study studied only one type of family -- one that's defined as having one or more children (and living with both parents, presumably).  Implications of the study's inclusion/exclusion criteria should be explored further.   

Since respondents were only those who worked in person, the model's findings might not hold for those who worked remotely, correct?  

p. 7: The variables sported approximately normal univariate distributions (Table 1).  More important for maximum likelihood estimation, though, would be whether they met the *multivariate* normality assumption.     

Author Response

Reviewer 2

Thank you for the opportunity to review this timely manuscript.  I believe that it would make a nice addition to the literature on this topic.  I provide a few suggestions and questions below:

p. 2: What does "social recreation" refer to in the context of this sentence?

Response 1. Thank you for the comment. We added a few words on that.

The authors appear to use "work exhaustion" and "emotional exhaustion" interchangeably when discussing this concept within their theoretical model, but readers may find that confusing. 

Response 2. Thank you for this comment. It gave us the opportunity to be consistent in the use of “Work exhaustion”. 

  1. 5: Were survey respondents drawn from the same workplace or company?   A "rich mix" of companies?  Were they recruited from a particular sector of the economy?  Did they work in fields/jobs that could be characterized by high levels of technostress?  Did they work in the private or public sector?  More information about their workplaces (and job roles, for that matter) would be helpful to contextualize these findings within the broader literature.    

Response 3. Thank you for this suggestion. We added all information that we gathered about our sample.

In terms of potential generalizability, the study studied only one type of family -- one that's defined as having one or more children (and living with both parents, presumably).  Implications of the study's inclusion/exclusion criteria should be explored further.   

Since respondents were only those who worked in person, the model's findings might not hold for those who worked remotely, correct?  

Response 4. Thank you for these comments. We added limitations and implications about the generalizability of the study.

  1. 7: The variables sported approximately normal univariate distributions (Table 1).  More important for maximum likelihood estimation, though, would be whether they met the *multivariate* normality assumption.     

Response 5: Thank you for this suggestion. We added a multivariate normality test which suggested that the assumption is not met. Therefore, we changed the estimation method of CFA and SEM from ML to MLR to adopt robust standard errors. We moved the paragraph concerning normality assumptions to the Data analysis section to explain the choice of the adoption of the MLR approach. Thus, we ran and present the analyses using MLR.

Reviewer 3 Report

Dear authors,

I have some suggestions regarding the conception of the paper and some minor remarks.

First the methodological / conceptional issue: I think the Paper would profitate of an more balanced argumentation. It is very strictly connected to the argument that there is negative Technostress associated with digital labour, that every parent perceive it in the same way,  that Home-Office is involuntary. 

Even if the home-office or working remotely have been involuntary during the pandemic, some parents could however perceive it as not more stressful. Other studies show also that they highlight the advantage of not commuting to work, flexible working hours to combine family and work tasks more souverign. Further in many studies there are inconsistencies already displayed: digital labour is both: it creates new sources of stress and in the same way helps solving others. By only focussing on the negative impacts these inconsistencies or also positive aspects are you not able to detect and to show. I think this needs attention / explanation or better a revision.

Also the items and indicators are selected in that way. If the paper keeps like this, then the results are not surprising as the methodological and theoretical assumptions does not allow any other results then the assumptions are made in the beginning.

Some other aspects:

- do fathers and mothers perceive "technostress" in the same way? Are there maybe any differences between the genders, because of their working duties?

- dual earner workers are not only vulnerable, they are also sometimes the most priviledged in the society. This is a not really clear group description because it depends on the jobs and how much they earn and their qualification, please give notice to that

Author Response

Reviewer 3

Dear authors,

I have some suggestions regarding the conception of the paper and some minor remarks.

First the methodological / conceptional issue: I think the Paper would profitate of an more balanced argumentation. It is very strictly connected to the argument that there is negative Technostress associated with digital labour, that every parent perceive it in the same way,  that Home-Office is involuntary. 

Even if the home-office or working remotely have been involuntary during the pandemic, some parents could however perceive it as not more stressful. Other studies show also that they highlight the advantage of not commuting to work, flexible working hours to combine family and work tasks more souverign. Further in many studies there are inconsistencies already displayed: digital labour is both: it creates new sources of stress and in the same way helps solving others. By only focussing on the negative impacts these inconsistencies or also positive aspects are you not able to detect and to show. I think this needs attention / explanation or better a revision.

Also the items and indicators are selected in that way. If the paper keeps like this, then the results are not surprising as the methodological and theoretical assumptions does not allow any other results then the assumptions are made in the beginning.

Response 1: Thank you for these comments. They gave us the opportunity to modify the abstract and to enrich the introduction with a more balanced argumentation about the positive aspects and consequences of digital transformation. Moreover, we better explained the variance of Technostress in our sample, underlining that some participants showed low levels of Technostress. It means that in answering to certain items, those participants declared that they do not perceive technology as a source of stress or negative outcomes. We interpreted this variability pointing out that some participants had positive experiences and some others had negative experiences with technologies. Therefore, we better explained that the positive relationship between Technostress and WF conflict suggests also that participants who had positive experiences with technology (i.e., low levels of Technostress) are those with less conflict between work and family, suggesting that a “good” use of technologies has positive consequences for WF dynamics.

Some other aspects:

- do fathers and mothers perceive "technostress" in the same way? Are there maybe any differences between the genders, because of their working duties?

Response 2. Thank you for these questions. They gave us the opportunity to add in the Results section that in our sample there are no gender differences in the perceptions of Technostress, WF balance, and Work exhaustion.

- dual earner workers are not only vulnerable, they are also sometimes the most priviledged in the society. This is a not really clear group description because it depends on the jobs and how much they earn and their qualification, please give notice to that

Response 4. Thank you for this comment. We added information about family incomes to better describe the characteristics of our sample.
